# The Chilean burden of disability-adjusted life years due to cardiovascular diseases: Results from the Global Burden of Disease Study 2021

Camilo Briones-Valdivieso[1], Claudia Nuñez[1], Andrés Celis[2], Jaqueline Araneda[3], Carlos Cristi-Montero[4], Gary O'Donovan[5], Carlos Celis-Morales[6,7,8], Salil V. Deo[9,10], Fanny Petermann-Rocha[6,11]*

**1** Facultad de Medicina, Universidad Diego Portales, Santiago, Chile, **2** Facultad de Odontología, Universidad de los Andes, Santiago, Chile, **3** Facultad de Ciencias de la Salud y de los Alimentos, Universidad del Bío-Bío, Chile, **4** IRyS Group, Physical Education School, Pontificia Universidad Católica de Valparaíso, Viña del Mar, Chile, **5** Facultad de Medicina, Universidad de los Andes, Bogotá, Colombia, **6** School of Cardiovascular and Metabolic Health, University of Glasgow, Glasgow, United Kingdom, **7** Human Performance Lab, Education, Physical Activity, and Health Research Unit, Universidad Católica del Maule, Talca, Chile, **8** High-Altitude Medicine Research Centre (CEIMA), Universidad Arturo Prat, Iquique, Chile, **9** Surgical Services, Louis Stokes Cleveland VA Medical Center, Cleveland, United States of America, **10** Case School of Medicine, Case Western Reserve University, Cleveland, United States of America, **11** Centro de Investigación Biomédica, Facultad de Medicina, Universidad Diego Portales, Santiago, Chile

* fanny.petermann@udp.cl

## Abstract

Cardiovascular disease (CVD) remains a leading cause of death and disability worldwide, including Chile. While mortality rates from CVD are well-documented, the associated burden of disability-adjusted life years (DALYs) and the impact of key modifiable risk factors have yet to be fully explored. This study aims to describe the temporal trends of CVD-related DALYs in Chile, stratified by gender, and to identify the primary risk factors contributing to this burden. An ecological study was conducted using data from the Global Burden of Disease Study 2021. We analysed age-standardised DALYs rates for overall and specific CVDs in Chile from 1990 to 2021. For the year 2021, we calculated the percentage distribution of DALYs by CVD type. Where applicable, CVDs were examined in relation to behavioural, metabolic, and environmental risk factors, and relative changes in DALYs over time. Over the last three decades, the overall CVD DALYs rate in Chile decreased substantially. Ischaemic heart disease and stroke accounted for most of the CVD burden, with gender differences observed. Stroke predominated in females and ischaemic heart disease in males. Metabolic risk factors, particularly high systolic blood pressure (SBP), were the most significant contributors to CVD DALYs, followed by behavioural and environmental risk factors. Although the rates of CVD DALYs have declined significantly in Chile during the past three decades, the burden remains substantial and with gender-specific differences. Metabolic risk factors, especially high SBP, followed by behavioural and

**Data availability statement:** Data are fully available at https://vizhub.healthdata.org/gbd-results/.

**Funding:** The author(s) received no specific funding for this work.

**Competing interests:** The authors have declared that no competing interests exist.

environmental factors, remain key contributors to CVD, highlighting the need for continued public health efforts focused on multi-level interventions to reduce the impact of these risk factors on cardiovascular health, such as adopting lower blood pressure goals among older people.

## Introduction

Cardiovascular disease (CVD) continues to be one of the leading causes of global mortality and disability worldwide [1]. The World Health Organisation has estimated that around 17.9 million people die yearly from CVD [2]. Although both high and low-middle-income countries (LMIC) have high rates of cardiovascular mortality, pre-mature cardiovascular mortality rates are higher among LMICs [3].

Smoking, physical inactivity, an unhealthy diet, elevated blood pressure, and poor glycaemic control are widely recognised modifiable risk factors contributing to CVD mortality [2]. However, apart from mortality, these risk factors also drive the increase in disability-adjusted life years (DALYs) attributed to CVD.

In Chile, cancer and CVD remain the leading contributors to the rising burden of DALYs. Although cancer continues to be the primary cause of mortality in this country, a previous study conducted during the COVID-19 pandemic demonstrated an alarming increase in CVD mortality, particularly among Chilean females [4]. While the global burden of CVD has been extensively documented, country-specific analyses are essential to inform local policy and intervention strategies. In Chile, the substantial impact of CVD is well-documented, with evidence from both mortality and DALYs metrics. However, the specific contribution of primary risk factors to this burden has yet to be fully investigated. The Global Burden of Disease (GBD) Study provides considerable information about morbidity and mortality at regional and global levels. However, its findings may not always translate directly into actionable insights for national-level decision-making. In Chile, for example, there are great differences in income, health, and healthcare compared to other Latin American countries [5]. Therefore, the main objective of this study is to examine the burden of CVD in Chile by analysing the magnitude and temporal trends of CVD DALYs, both overall and those attributable to specific risk factors. The analysis includes all CVDs available in the GBD database, emphasising those for which risk factor data are available.

## Methods

### Data source

The GBD is an ecological research study that systematically quantifies the burden of various diseases and risk factors across different countries by utilising diverse data sources, including published studies, household surveys, censuses, administrative data, ground monitor data, or remote sensing [6]. GBD provides standardised epide-miological data and summary health measures, such as DALYs used for this study [7]. The complete GBD methodology for estimating the CVD burden due to specific

risk factors can be found elsewhere [8]. Data were obtained from the GBD 2021 edition via the online Global Health Data Exchange query tool (GHDx, http://ghdx.healthdata.org/gbd-results-tool) [9].

### Disability-adjusted life years

DALYs are an epidemiological measure that provides a numerical estimate regarding the total years of healthy life lost due to specific causes and risk factors at the population level. Each DALY is equivalent to one year of life lost to illness or disability and was defined as the sum of years of life lost (YLL) and years lived with disability (YLD). DALYs were estimated from life tables, estimates of prevalence, and disability weights. They are expressed as rates per 100,000 Chilean individuals, with a 95% uncertainty interval (UI).

### Cardiovascular diseases and risk factors

By utilising their DALYs rate, the following CVDs were included in the analysis: overall CVD, atrial fibrillation and flutter, aortic aneurysm, cardiomyopathy and myocarditis, endocarditis, hypertensive heart disease, ischaemic heart disease, lower extremity peripheral arterial disease (PAD), non-rheumatic valvular heart disease, pulmonary arterial hypertension, rheumatic heart disease, stroke (overall, including ischaemic and haemorrhagic), and other cardiovascular and circulatory diseases.

Data on DALYs rates attributed to risk factors were available for overall CVD, atrial fibrillation and flutter, aortic aneurysm, cardiomyopathy and myocarditis, hypertensive heart disease, ischaemic heart disease, PAD, and stroke. The GBD study organised the risk factors by level; the first-level risk factors are i) behavioural, ii) environmental/occupational, and iii) metabolic. The second-level risks were derived from the first-level, and all of them were considered; however, the ones with data for the selected diseases were: i) derived from behavioural risks: alcohol use, dietary risks, low physical activity and tobacco; ii) derived from environmental/occupational risks: air pollution, non-optimal temperature, other environmental risks; iii) derived from metabolic risks: high body-mass index (BMI), high fasting plasma glucose (FPG), high low-density lipoprotein (LDL) cholesterol, high systolic blood pressure (SBP), and kidney dysfunction. Third-level risk factors, such as those derived from dietary risks, were not included in this analysis in order to enhance clarity and interpretability, and to focus on the most impactful and policy-relevant contributors to the CVD burden.

### Statistical analyses

From GHDx [9], the age-standardised DALYs rate (overall and by gender), attributed to each CVD and the aforementioned risk factors, was obtained. The percentage distribution of DALYs for each CVD in 2021 was calculated as the proportion relative to the overall CVD DALYs rate. To evaluate the relative changes in DALYs over time, the relative percentage change in DALYs over the study period was calculated as follows:

$$\left[\frac{age\ adjusted\ DALY\ (2017-2021) - age\ adjusted\ DALY\ (1990-1994)}{age\ adjusted\ DALY\ (1990-1994)}\right] * 100$$

The 95% confidence interval (CI) for the relative change was calculated using the delta method. To describe changes over time in the relative importance of each risk factor for each cause of death, rankings were created to compare the evolution of these risk factors. Analyses were performed in R (version 4.3.1). As this study is based on descriptive analyses of age-standardised DALYs rates and GBD-derived estimates, no additional statistical tests were applied to compare gender differences. Interpretations are based on visual and descriptive comparisons within reported UI.

### Ethics considerations

This study used aggregated and publicly available data from the GBD Study; therefore, no ethical approval was required.

   

## Results

### Overall CVD

As illustrated in Fig 1, the CVD DALYs rate has continuously decreased over the years in both genders. The overall CVD DALYs rate decreased from 4961.2 (95% UI: 4764.4; 5114.4) in 1990 to 2202.1 (95% UI: 2061.4; 2341.0) in 2021. The percentage distribution of DALYs for CVDs in 2021 is depicted in Fig 2, where is shown that 73.1% of total DALYs were from both ischaemic heart disease and Stroke. Fig 3 highlights that the stroke rate for females in 2021 was 690.0 (95% UI: 624.4; 750.2), representing 41.6% of the total female CVD DALYs rate , while ischaemic heart disease accounted for 28.4%. Conversely, among males, the DALYs rate for ischaemic heart disease was 1200.2 (95% UI: 1139.0; 1261.7), representing 42.3% of the total male DALYs rate, while stroke accounted for 33.0%.

### CVD over the years

The relative change in CVD DALYs rates from 1990–1994–2017–2021 is depicted in Fig 4, where is shown that most CVDs exhibited a decreased rate over the years, particularly rheumatic heart disease, which decreased by 81.1%. However, atrial fibrillation and flutter, endocarditis, and other cardiovascular and circulatory diseases DALYs rates increased by 7.4% (95% CI: 3.9; 11.0), 41.9% (95% CI: 32.9; 50.8), and 70.6% (95% CI: 65.5; 75.8), respectively, for the whole Chilean

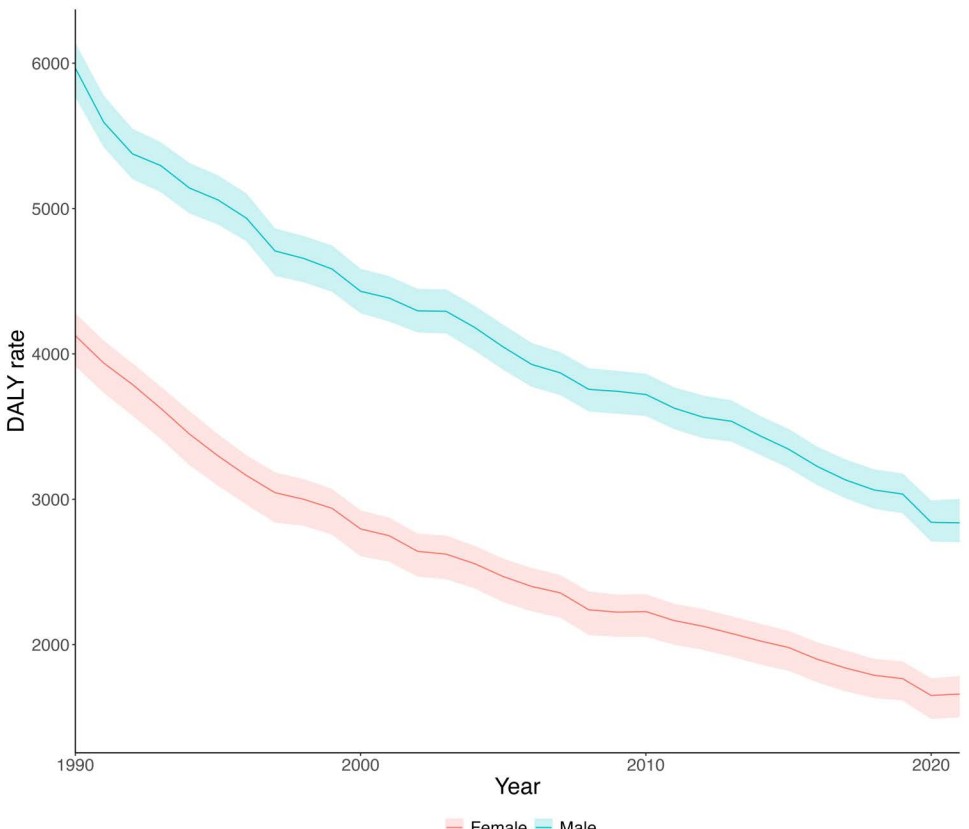

**Fig 1. Overall cardiovascular disease DALYs age-standardised rate over time 1990-2021 in Chile.** DALYs rates are expressed per 100,000 individuals, with a 95% uncertainty interval. The overall CVD DALYs rate has decreased in both females and males over the years.

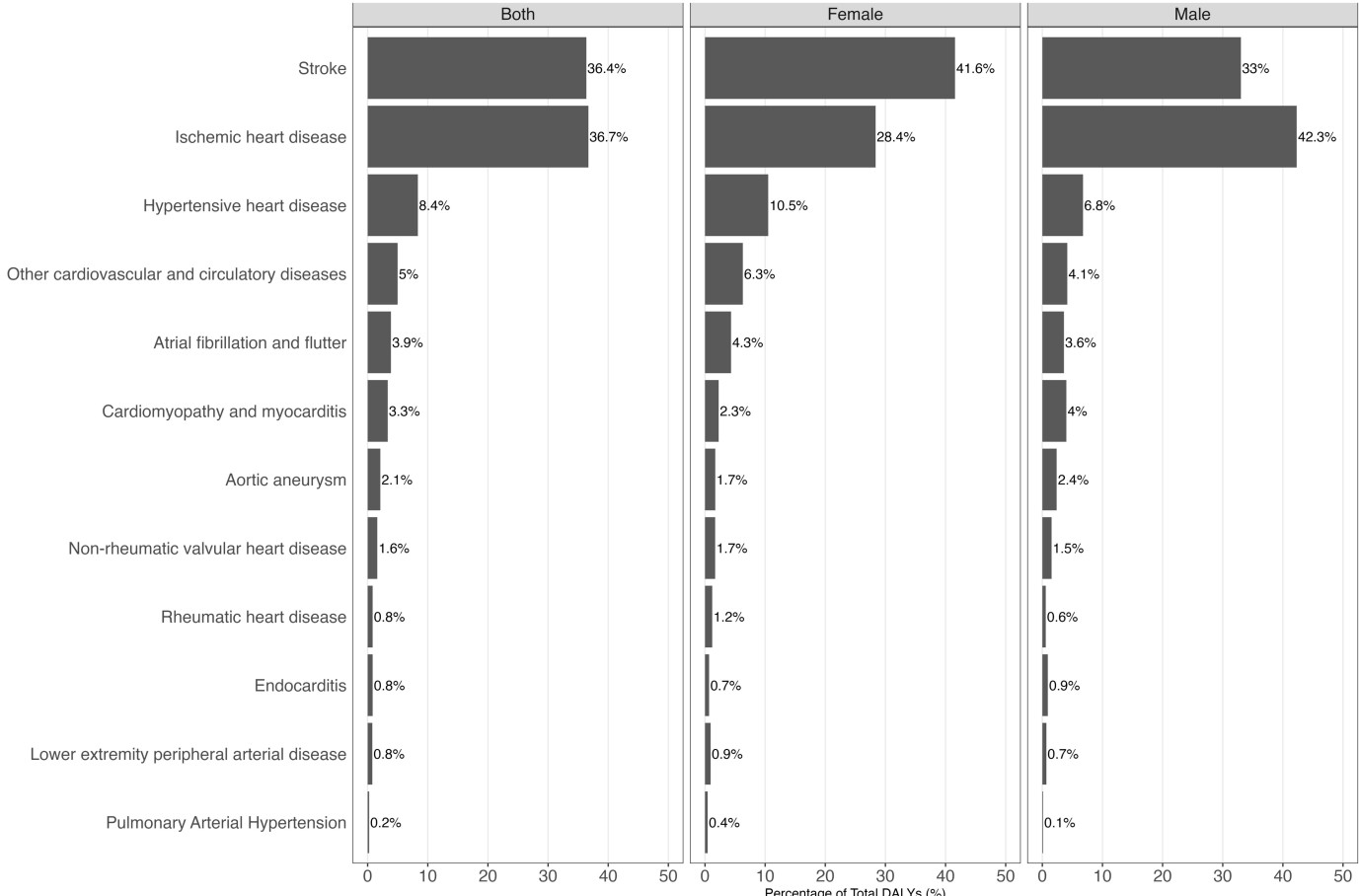

**Fig 2. Percentage distribution of DALYs for cardiovascular diseases in 2021 in Chile.**

population. As shown in Fig 3, the relative importance has changed over the years. For instance, stroke, ischaemic heart disease, and hypertensive heart disease remained as the leading diseases. However, rheumatic heart disease stood out among females and males, moving from the 4th to the 9th and from the 5th to the 11th place, respectively, while other cardio-vascular and circulatory diseases from the 7th to the 4th in both genders. Also, among males, endocarditis moved from the 11th to the 9th position.

## CVD risk factors

The overall CVD disease burden in DALYs attributable to the 1st-level risk factors is shown in Table 1, highlighting that – when comparing first-level risk factors over time – metabolic risk factors consistently accounted for the highest overall CVD-DALYs. In 2021, this rate was 1468.9 (95% UI: 1319.2; 1615.4) for both genders. The latter is followed by behavioural risks (851.3 [95% UI: 533.4; 1053.4]) and environmental/occupational factors (476.0 [95% UI: 377.2; 588.4]). The absolute and relative overall CVD DALYs rate attributable to the 1st-level risk factors is depicted in Fig 5, showing a steady decline in all categories over time, with relative changes between 1990−1994 and 2017−2021 of −50.6% (95% CI: −56.6; −44.7) for metabolic risks, −55.5% (95% CI: −62.1; −48.9) for behavioural risks, and −63.7% (95% CI: −71.7; −55.7) for environmental/occupational risks. At the second-level risk factor

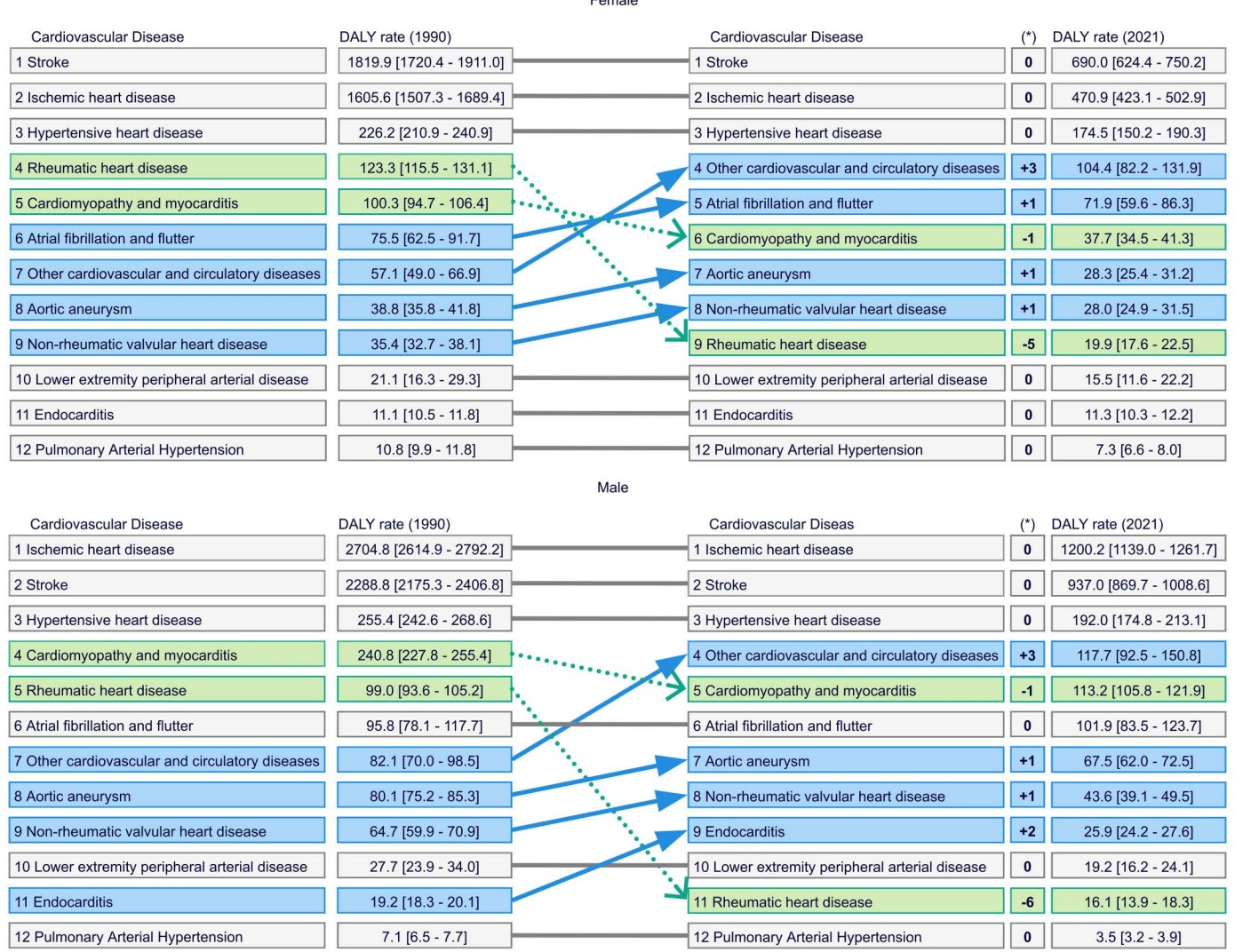

**Fig 3. Cardiovascular diseases ranked by DALYs in 1990 and 2021 for females (above) and males (below) in Chile.** Changes in relative importance, measured by differences in ranking position, are highlighted. CVDs with a decrease in relative importance are shown in green, while those with an increase are shown in blue.

analysis, absolute and relative changes over the same period are shown in Fig 6. High SBP and dietary risks were consistently the leading risk factors for both females and males from 1990 to 2021 (Fig 7). For both genders, the 2021 high SBP DALYs rate was 1134.1 (95% UI: 949.6; 1292.4), and for dietary risks, it was 587.3 (95% UI: 214.3; 821.4). High BMI rose to the 3rd place among females, with a DALYs rate of 291.5 (95% UI: 160.2; 447.1), while high LDL cholesterol held that spot among males with a rate of 621.4 (95% UI: 399.8; 838.0). Notably, as shown in Fig 7, where the overall CVD 2nd level risk factors rank ordered by attributed DALYs in 1990 and 2021 is exhibited, air pollution, the leading environmental/occupational risk, dropped from the 3rd to the 5th in females and from the 3rd to the 6th in males. In contrast, low physical activity consistently had the lowest DALYs rate among all risk factors throughout the years.

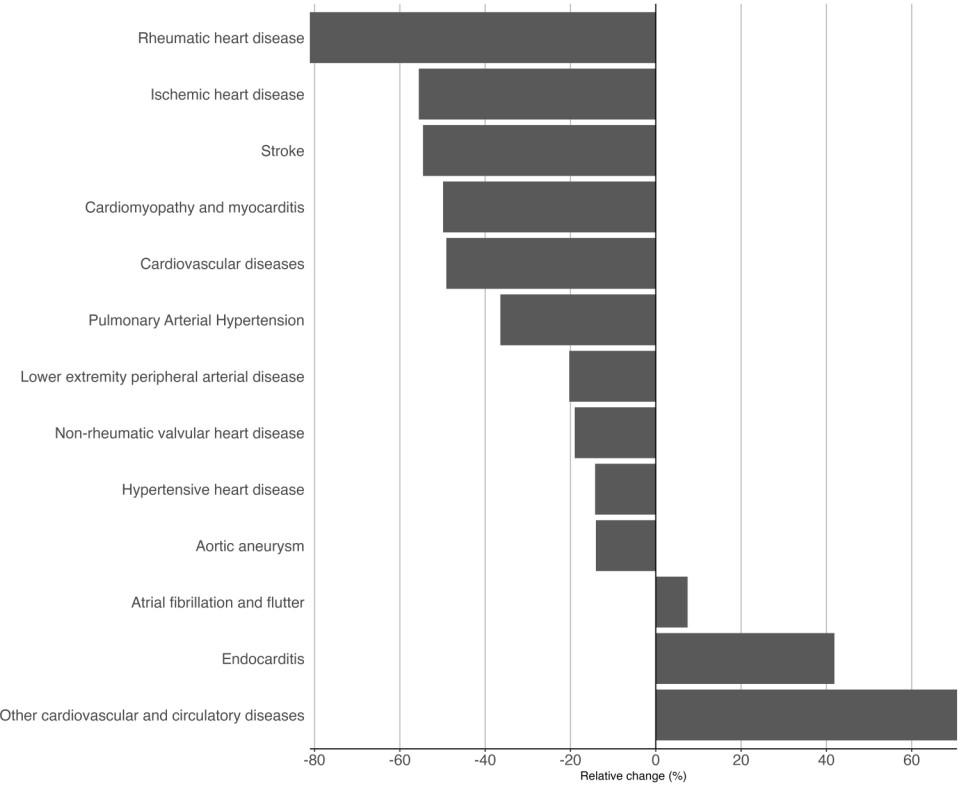

**Fig 4. Relative change in cardiovascular diseases DALYs rates from 1990-1994 to 2017-2021 in Chile, for both genders.** Rheumatic heart disease had the most significant relative decrease, and other cardiovascular and circulatory diseases showed the biggest relative increase.

**Table 1. Overall CVD disease burden in DALYs attributable to 1st-level risk factors in Chile. DALYs rates are expressed per 100,000 individuals, with a 95% uncertainty interval.**

| Risk Factor | DALYs Rate per 100,000 individuals | | | | | |
|---|---|---|---|---|---|---|
| | **Both** | | **Female** | | **Male** | |
| | **1990** | **2021** | **1990** | **2021** | **1990** | **2021** |
| Metabolic risks (overall) | 3414.6 [3054.1-3714.2] | 1468.9 [1319.2-1615.4] | 2855.3 [2542.7-3127.5] | 1076.4 [928.1-1198.7] | 4083.4 [3629.9-4450.7] | 1928.6 [1731.2-2098.2] |
| Behavioural risks | 2222.3 [1344.1-2759.1] | 851.3 [533.4-1053.4] | 1653.5 [973.6-2094.7] | 532.2 [341.4-673.5] | 2902.5 [1771.1-3559.5] | 1220.2 [760.9-1503.7] |
| Environmental/occu-pational risks (overall) | 1558 [1238.1-1889.7] | 476 [377.2-588.4] | 1287.7 [1026.5-1553.4] | 340.2 [266.2-423.2] | 1883.5 [1484.9-2292.6] | 635.3 [501.6-785.1] |

The S1-S7 Figs. show that DALYs rates attributable to the second-level risk factors for ischaemic heart disease, stroke, hypertensive heart disease, cardiomyopathy and myocarditis decreased across most risks between 1990–1994 and 2017–2021. However, for atrial fibrillation and flutter, the DALYs rate attributed to high BMI increased by 71.2% (95% CI: 68.2; 74.3) for both genders, alongside increases in other environmental risks (28.6% [95% CI: 24.5; 32.7]) and high SBP (7.7% [95% CI: 4.1; 11.3]). For PAD, the DALYs rate associated with high FPG rose by 45.5% (95% CI: 39.7; 51.2), while the rate attributed to high BMI increased by 5.3% (95% CI: 0.7; 9.9). For aortic aneurysm, high BMI led to a 7.4% (95% CI: 4.7; 10.2) rise in DALYs rates.

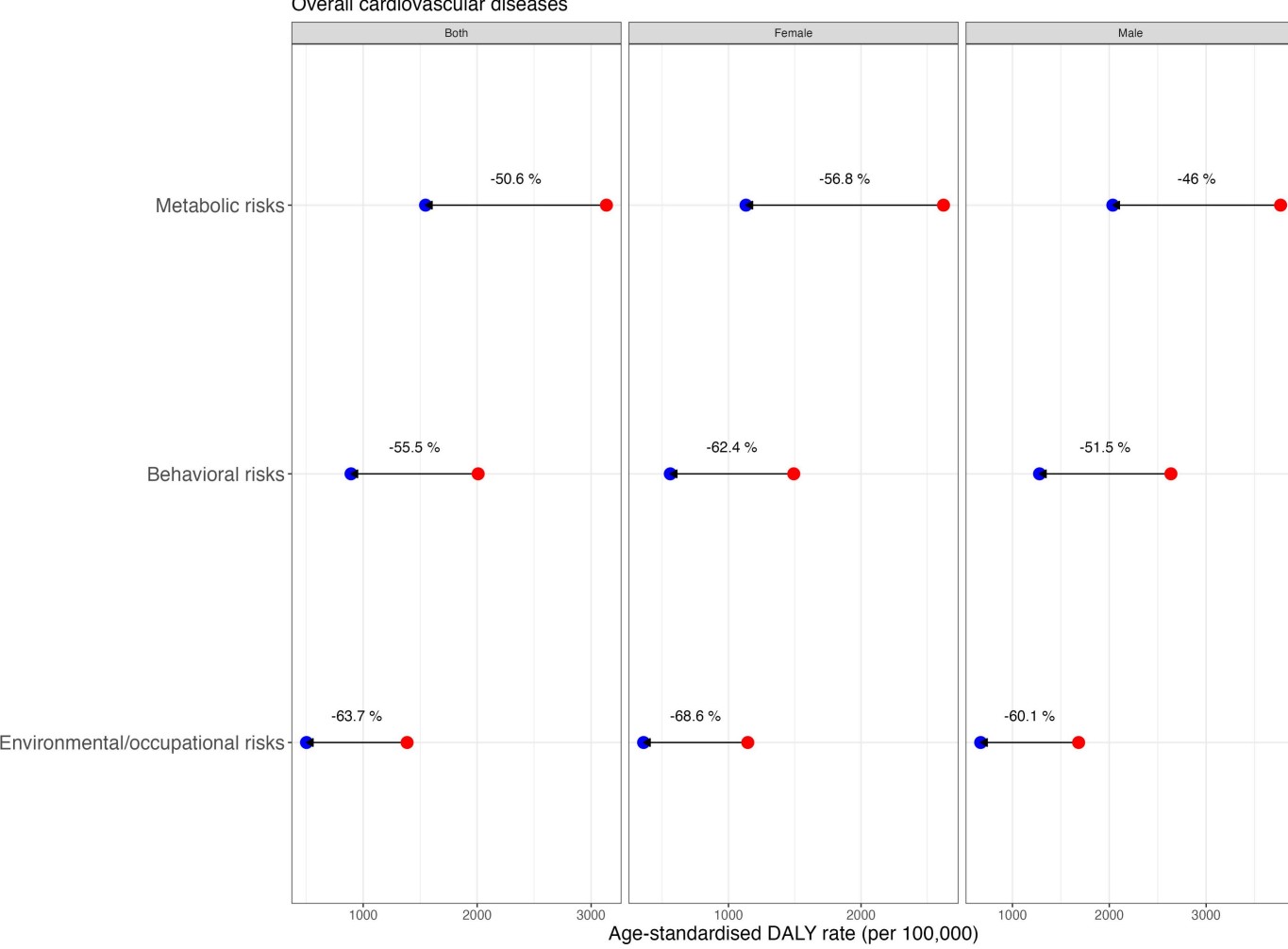

**Fig 5. Absolute and relative overall CVD DALYs rate attributable to 1ˢᵗ-level risk factors in Chile.** The red dot represents the mean 5-year period for 1990-1994, and the blue dot represents the 2017-2021 period.

## Discussion

### Main findings

This study assessed the burden of CVD in Chile by analysing DALYs for both overall and specific conditions, as well as those attributable to risk factors. Stroke in females and ischaemic heart disease in males were identified as the primary contributors to DALYs rates, highlighting notable gender differences. Over the years, the overall DALYs rate for CVDs has declined, with rheumatic heart disease showing the most significant reduction. However, atrial fibrillation, flutter, and endocarditis have shown increasing trends. Metabolic risk factors, particularly high SBP, emerged as the leading contributors to the overall burden, followed by behavioural and environmental/occupational risk factors, all demonstrating a continuous decline over time. Nonetheless, some specific risk factors, such as high BMI, have increased their impact on certain conditions, including PAD, atrial fibrillation and flutter, and aortic aneurysm.

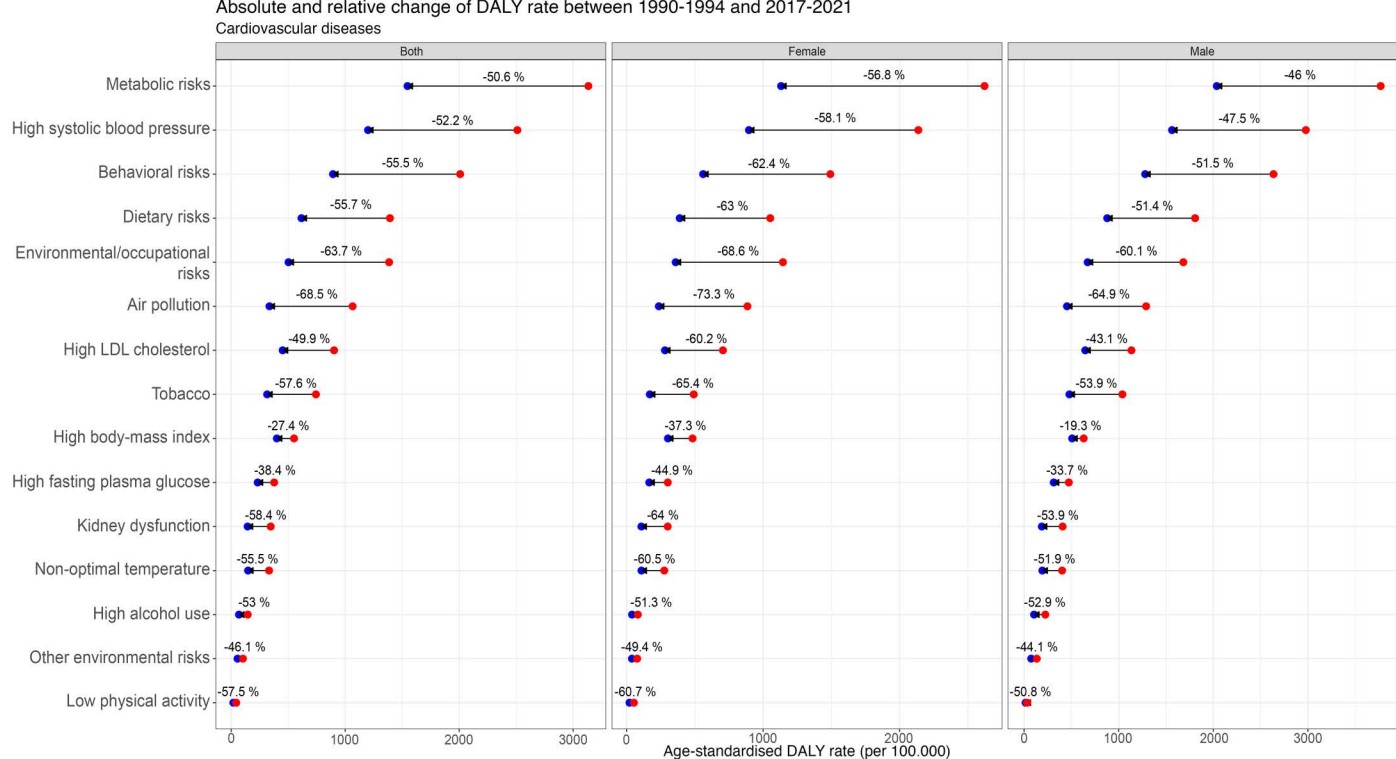

**Fig 6. Absolute and relative overall CVD DALYs rate attributable to 2nd-level risk factors in Chile.** The red dot represents the mean 5-year period for 1990-1994, and the blue dot represents the 2017-2021 period.

To the best of our knowledge, this is the first study to comprehensively assess the long-term trends in CVD DALYs and their associated risk factors in Chile. These findings hold significant implications for policy and practice, underscoring the need for targeted public health interventions to address the evolving burden of CVDs in the country.

## Interpretation and implications

As stroke and ischaemic heart disease have invariably been the leading CVDs in terms of DALYs rate worldwide, greater efforts to reduce their burden should be considered. The gender-specific distribution of these conditions—stroke in women and ischaemic heart disease in men—suggests that gender-sensitive strategies may be relevant in the context of primary prevention. However, given the substantial disability burden following acute cardiovascular events, novel approaches should be considered. For instance, tackling ischaemia–reperfusion injury could be particularly relevant in ischaemic stroke and acute myocardial infarction [10]; for the latter, myocardial reperfusion injury contributes up to 50% of the final myocardial infarct size [11]. In this regard, although antioxidants have been successfully tried in preclinical studies, translating positive experimental results into clinical models has been challenging. Thus, strategies based on multi-antioxidant therapies for myocardial infarction [10] and stroke [10,12] have been proposed.

Although DALYs and mortality rates from stroke have decreased in Latin America, absolute numbers are rising due to population ageing [13]. A Chilean study found low adherence to evidence-based performance measures for acute ischemic stroke in public hospitals in the capital, with less than 50% of patients receiving a computed tomography scan within 24 hours of symptom onset and fewer than 10% within 4.5 hours, limiting acute interventions [14]. Additionally, a national Chilean survey assessing stroke knowledge and awareness found that nearly 25% of participants failed to

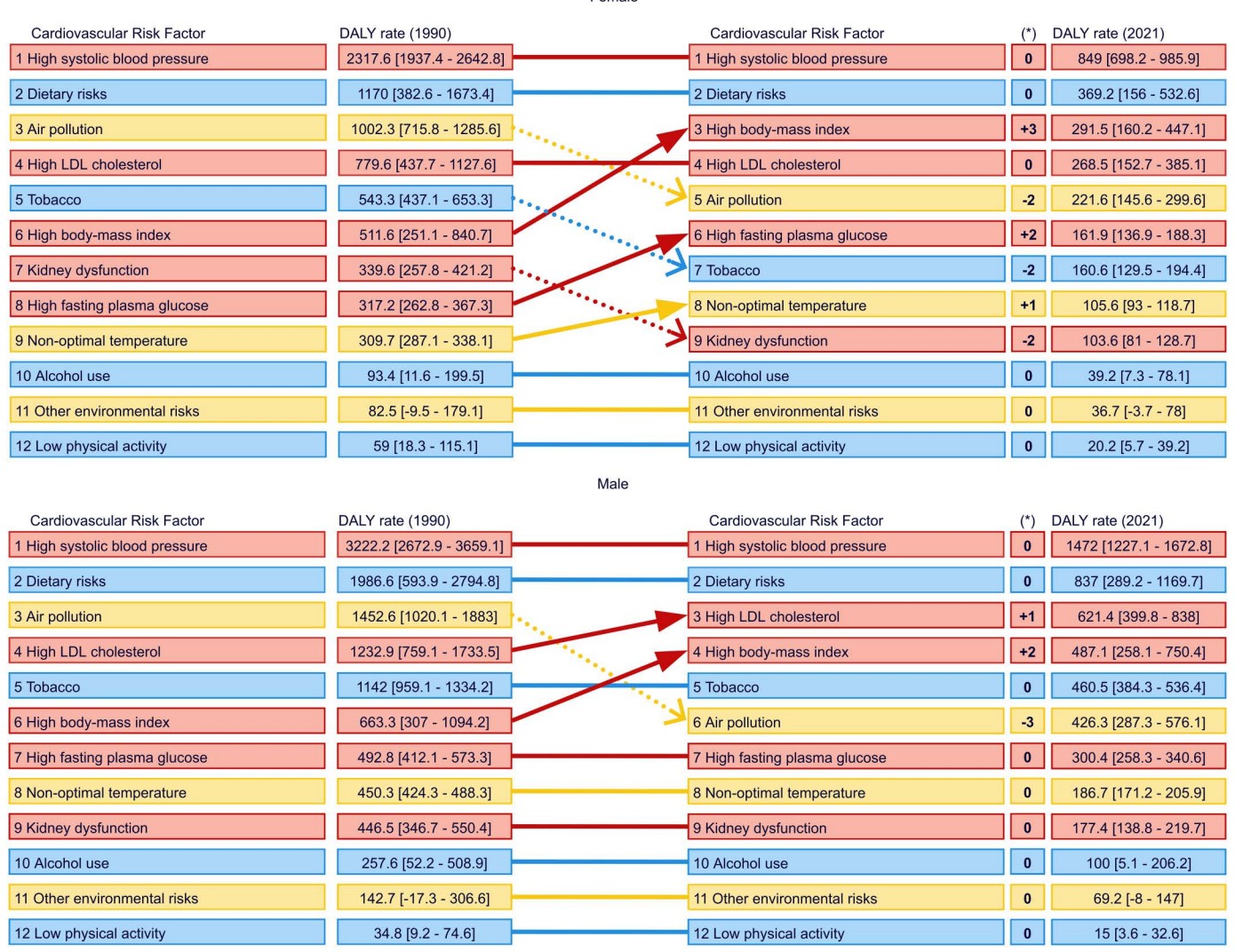

**Fig 7. Overall cardiovascular disease 2nd level risk factors rank ordered by attributed DALYs in 1990 and 2021 for females (above) and males (below) in Chile. Red.** Metabolic risks. Blue. behavioural risks. Yellow. environmental/occupational risks.

recognise at least one typical stroke symptom, and recognition of risk factors was low, with only 42% identifying hypertension as a risk [13].

The DALYs rate for atrial fibrillation and flutter has been increasing, particularly among males, and the burden attributed to high BMI has also risen significantly. Globally, atrial fibrillation and flutter are significant health issues, with prevalence increasing 1.37-fold over 31 years, which can impact quality of life. Interestingly, ageing has been proposed as the pivotal risk for atrial fibrillation over other factors [15]. This is particularly relevant due to the association between atrial fibrillation, cognitive impairment and dementia. Indeed, atrial fibrillation may independently increase the risk of vascular contributions to cognitive impairment and dementia (VCID), correlating with a higher incidence of dementia, even in the absence of clinically apparent strokes [16]. Thus, preventive and therapeutic measures may be considered, such as (i) early detection and prevention of VCID, with awareness campaigns, patient education, smart devices, and diagnosis of paroxysmal atrial fibrillation; (ii) anticoagulation therapy, which plays a crucial role in reducing stroke risk in atrial fibrillation patients; (iii) rate

and rhythm control; (iv) multifactorial approaches. Managing comorbid conditions that contribute to both atrial fibrillation and VCID, such as hypertension and diabetes, is vital. Hence, a multifactorial approach that includes lifestyle modifications is essential for reducing the risk of cognitive decline [16].

Metabolic risks and particularly high SBP emerged as the main leading risk factors for CVD burden, findings consistent with numerous studies of other regions of the world [6,17–19]. In this context, lower blood pressure goals have been proposed to decrease the incidence of cardiovascular events [20,21] with no significant differences in adverse events, such as symptomatic hypotension, injurious falls, and syncope, among old patients [21].

CVD burden attributable to dietary risk factors decreased by around 50% in the past decades; these risks emerged as the second leading risk factor for overall CVDs. Specific dietary risks were not assessed in the current analysis; however, a previous work analysing data from the Southern Latin America Region (Argentina, Chile and Uruguay) [17] described a decrease in the exposure to "low in" diets (whole grains, legumes, seafood omega-3 fatty acids, nuts and seeds, polyunsaturated fatty acids, vegetables, fibre and fruits) and increasing exposure to some "high in" diets (red meat, sugar-sweetened beverages, and processed meat), except for the decrease of exposure to diets high in trans fatty acids and sodium.

Although air pollution's relative importance as a CVD risk factor has decreased over the decades, its burden is still greater than other well-recognised risk factors, such as high FPG. Fine particulate matter <2.5 microns ($PM_{2.5}$) is particularly harmful, and no South American country achieved the WHO recommended limit [22]. Air pollution is not only damaging in the long term but also in short periods (hours or days), increasing the risk of myocardial infarctions, strokes, and CVD mortality [22]. In Chile, although the CVD DALYs rate attributed to air pollution decreased over the years, this effect is mainly due to a decrease in household air pollution; meanwhile, there is no significant improvement in the burden of ambient air pollution disease [23].

While our findings highlight critical areas for public health attention, more defining specific and context-based interventions for Chile would require additional studies incorporating local health system characteristics, implementation feasibility, and cost-effectiveness evaluations.

## Strengths and limitations

This study presents an updated overview of the CVD burden and the role of major risk factors, considering gender-related differences and focusing not only on recent data but also on the trends over the decades in Chile. Furthermore, focusing on one country made it possible to provide specific and useful information for physicians and policymakers.

Although the burden of other cardiovascular and circulatory diseases has increased over the years, limited information is available about these conditions. Similarly, this study does not include data on third-level risk factors, such as specific dietary risks.

Another limitation of this study lies in how low physical activity was measured within the GBD framework. Physical activity is a complex, multidimensional behaviour influenced by factors such as intensity, duration, type, and domain [24]. However, the GBD study adopted a simplified approach, using only a single indicator to represent low physical activity as a risk factor. This contrasts with the more detailed assessment of other behavioural risk factors, such as dietary risks, which were analysed using multiple specific indicators, giving rise to numerous third-level risk factors [24]. This oversimplification may underestimate the health risks associated with low physical activity levels and could lead to misunderstandings among health workers and policymakers regarding its true impact.

Additionally, the results of this study only report national-level data and do not contain any subnational information that may be relevant to assess the impact of cultural and behavioural differences within territories with completely different climates and influences from Indigenous communities [17]. Furthermore, it has been declared that the availability and quality of primary data used for the GBD estimates have been heterogeneous and considered a limitation by the GBD study itself [8]. Finally, as an ecological study, the ecological fallacy is always plausible.

## Conclusion

Over the decades, DALYs rates for most CVDs have declined in Chile, reflecting successful strategies in prevention and treatment; however, stroke, ischaemic heart disease and hypertensive heart disease continue to be the leading CVDs. Metabolic, behavioural, and environmental risk factors remain significant contributors to the CVD burden in Chile, with high SBP and dietary risks as the leading risk factors. This highlights the need for targeted multi-level interventions and comprehensive strategies to mitigate their impact on cardiovascular health.

## Supporting information

**S1 Fig. Absolute and relative ischaemic heart disease DALYs rate attributable to 2nd level risk factors in Chile.** The red dot represents the mean 5-year period for 1990–1994, and the blue dot represents the 2017–2021 period. (TIF)

**S2 Fig. Absolute and relative stroke DALYs rate attributable to 2nd level risk factors in Chile.** The red dot represents the mean 5-year period for 1990–1994, and the blue dot represents the 2017–2021 period. (TIF)

**S3 Fig. Absolute and relative hypertensive heart disease DALYs rate attributable to 2nd level risk factors in Chile.** The red dot represents the mean 5-year period for 1990–1994, and the blue dot represents the 2017–2021 period. (TIF)

**S4 Fig. Absolute and relative lower extremity peripheral arterial disease DALYs rate attributable to 2nd level risk factors in Chile.** The red dot represents the mean 5-year period for 1990–1994, and the blue dot represents the 2017–2021 period. (TIF)

**S5 Fig. Absolute and relative aortic aneurysm DALYs rate attributable to 2nd level risk factors in Chile.** The red dot represents the mean 5-year period for 1990–1994, and the blue dot represents the 2017–2021 period. (TIF)

**S6 Fig. Absolute and relative atrial fibrillation and flutter DALYs rate attributable to 2nd level risk factors in Chile.** The red dot represents the mean 5-year period for 1990–1994, and the blue dot represents the 2017–2021 period. (TIF)

**S7 Fig. Absolute and relative cardiomyopathy and myocarditis DALYs rate attributable to 2nd level risk factors in Chile.** The red dot represents the mean 5-year period for 1990–1994, and the blue dot represents the 2017–2021 period. (TIF)

## Contributors

CB-V and FP-R have conceptualised and designed the study. CB-V performed the analyses and wrote the first draft with the support of FP-R. All authors critically reviewed this and previous drafts. All authors approved the final draft for submission, with final responsibility for publication. FP-R is the guarantor of this paper.

## Author contributions

**Conceptualization:** Camilo Briones-Valdivieso, Salil V Deo, Fanny Petermann-Rocha.

**Formal analysis:** Camilo Briones-Valdivieso.

**Investigation:** Fanny Petermann-Rocha.

**Project administration:** Fanny Petermann-Rocha.

**Supervision:** Salil V Deo, Fanny Petermann-Rocha.

**Writing – original draft:** Camilo Briones-Valdivieso.

**Writing – review & editing:** Claudia Nuñez, Andrés Celis, Jaqueline Araneda, Carlos Cristi-Montero, Gary O'Donovan, Carlos Celis-Morales, Salil V Deo, Fanny Petermann-Rocha.

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
