## [Decision Letter · Decision Letter 0]

4 Mar 2025

PONE-D-24-57148The Chilean Burden of Disability-Adjusted Life Years due to Cardiovascular Diseases: Results from the Global Burden of Disease Study 2021PLOS ONE

Dear Dr. Petermann-Rocha,

Thank you for submitting your manuscript to PLOS ONE. After careful consideration, we feel that it has merit but does not fully meet PLOS ONE’s publication criteria as it currently stands. Therefore, we invite you to submit a revised version of the manuscript that addresses the points raised during the review process.

We look forward to receiving your revised manuscript.

Kind regards,

Amir Hossein Behnoush

Academic Editor

PLOS ONE

Journal Requirements:

2. Thank you for stating the following in your Competing Interests section: “None to declare”

Reviewers' comments:

Reviewer's Responses to Questions

**Comments to the Author**

1. Is the manuscript technically sound, and do the data support the conclusions?

Reviewer #1: Yes

Reviewer #2: Yes

Reviewer #3: Yes

2. Has the statistical analysis been performed appropriately and rigorously? 

Reviewer #1: No

Reviewer #2: Yes

Reviewer #3: Yes

3. Have the authors made all data underlying the findings in their manuscript fully available?

Reviewer #1: Yes

Reviewer #2: Yes

Reviewer #3: Yes

4. Is the manuscript presented in an intelligible fashion and written in standard English?

Reviewer #1: Yes

Reviewer #2: Yes

Reviewer #3: Yes

5. Review Comments to the Author

Reviewer #1: Abstract

• The abstract lacks a mention of statistical methods used in the study. Please include a brief description to enhance clarity and transparency.

• The abstract does not provide concrete recommendations for public health or policymaking. Incorporate actionable outcomes to strengthen its impact and relevance.

Introduction

• The transition from global statistics to Chile-specific challenges is abrupt, disrupting the narrative flow. Include a bridging sentence to improve coherence, such as: "While the global burden of CVD is well-documented, Chile faces unique challenges stemming from its diverse geography, aging population, and healthcare disparities."

• Repeated phrases like "leading cause of mortality". Consolidate by using synonyms, such as replacing "leading cause of mortality" with "primary contributor to death and disability," to enhance readability and avoid redundancy.

Methods

• The gender-specific analysis methodology is not described adequately. It is unclear whether statistical tests were applied to validate differences.

• Excluding third-level risk factors is not sufficiently justified, which may leave readers questioning the comprehensiveness of the study.

• No mention of ethical approvals or permissions for using secondary data, which is a standard requirement for publication.

• Include statistical tests used for gender-specific trends, e.g., "A t-test was conducted to evaluate the statistical significance of differences between gender-specific DALYs."

• Justify exclusions, e.g., "Third-level risk factors were excluded to simplify analysis and focus on the most impactful contributors to CVD burden."

• Add a note on ethical considerations, e.g., "As this study utilized publicly available secondary data, no ethical approval was required."

Results

• Figures and tables are referenced but not thoroughly explained in the text, which may confuse readers unfamiliar with the dataset.

• Trends like the increase in atrial fibrillation and endocarditis are mentioned but not explored in depth.

• Add explanations for key figures, e.g., "Figure 1 illustrates the decline in CVD DALYs over time, with the steepest reduction observed between 2000 and 2010."

• Analyse trends more deeply, e.g., "The rise in atrial fibrillation and flutter may be attributed to increased detection rates due to advancements in diagnostic tools."

Discussion

• The discussion about atrial fibrillation and vascular cognitive impairment is lengthy but loosely connected to the study’s objectives.

• Recommendations are broad and lack specificity for Chile.

• Focus on directly relevant implications, e.g., "The gender-specific burden of stroke and ischemic heart disease highlights the need for tailored interventions addressing unique risk profiles in men and women."

• Provide concrete recommendations, e.g., "Increased funding for hypertension screening programs in rural areas of Chile could reduce DALYs attributed to high systolic blood pressure."

Reviewer #2: It is better to stratify not only by sex, but also by other very important variables; for instance; residence, occupation et. other wise the paper looks nice; on the my view it is interesting and pass to publication.

Reviewer #3: The manuscript titled "The Chilean Burden of Disability-Adjusted Life Years due to Cardiovascular Diseases: Results from the Global Burden of Disease Study 2021" is a research study that examines the impact of cardiovascular diseases (CVD) on the population of Chile over the past three decades. The study uses data from the Global Burden of Disease Study 2021 to analyze age-standardized rates of disability-adjusted life years (DALYs) due to various CVDs from 1990 to 2021, focusing on gender differences and key risk factors.

The study highlights the need for continued public health efforts and multi-level interventions to address the key risk factors contributing to CVD in Chile. The authors emphasize the importance of targeted strategies to reduce the impact of high SBP, dietary risks, and other modifiable risk factors on cardiovascular health. I want to congratulate the authors on embarking on this all-important study.

The methodology used in the study appears to be robust and comprehensive.

However, this study has some limitations. As with any ecological research, there is a risk of ecological fallacy, where associations observed at the population level may not necessarily apply to individuals. The study notes that the measurement of low physical activity within the GBD framework is simplified, which may underestimate its impact. The study reports national-level data and does not include subnational information, which could be relevant for understanding regional differences within Chile. Overall, the methodology is well-suited to the study's objectives and thoroughly analyses the CVD burden and its risk factors in Chile.

While the study is comprehensive and well-conducted, several aspects could be improved to enhance its robustness and applicability.

Incorporating subnational data could provide more detailed insights into regional variations within Chile. This would help to identify specific areas with higher burdens of CVD and tailor public health interventions more effectively.

The study mentions that the measurement of low physical activity is simplified. A more detailed analysis that considers different dimensions of physical activity (e.g., intensity, duration, type, and domain) could provide a clearer understanding of its impact on CVD.

The study excludes third-level risk factors, such as specific dietary risks. Including these could offer a more nuanced understanding of how specific dietary components contribute to the CVD burden.

While the study examines trends over time, a more detailed longitudinal analysis could help to identify causal relationships and the impact of specific interventions over the study period.

Including socioeconomic factors in the analysis could provide insights into how income, education, and access to healthcare influence the CVD burden and its risk factors.

Incorporating qualitative data, such as patient interviews or focus groups, could provide context to the quantitative findings.

Analyzing the impact of specific public health policies and interventions implemented in Chile over the study period could help to identify successful strategies and areas needing improvement.

Expanding the range of environmental factors considered, such as urbanization, climate change, and occupational hazards, could provide a more comprehensive view of their impact on CVD.

6. PLOS authors have the option to publish the peer review history of their article (what does this mean? ). If published, this will include your full peer review and any attached files.

**Do you want your identity to be public for this peer review?** For information about this choice, including consent withdrawal, please see our Privacy Policy .

Reviewer #1: **Yes: ** Faizul Akmal Abdul Rahim

Reviewer #2: No

Reviewer #3: **Yes: ** U S H GAMAGE

---

## [Author Response · Author response to Decision Letter 0]

15 Apr 2025

Reviewer #1:

Abstract

1. The abstract lacks a mention of statistical methods used in the study. Please include a brief description to enhance clarity and transparency.

Response: We thank the reviewer for the comment. A brief description was included in the abstract, lines 34 to 39.

2. The abstract does not provide concrete recommendations for public health or policymaking. Incorporate actionable outcomes to strengthen its impact and relevance.

Response: Thank you for this valuable suggestion. While the primary aim of our study is to underscore the burden and trends of cardiovascular disease risk factors in Chile, we agree that highlighting actionable outcomes can enhance the relevance of the abstract. To address this, we have briefly mentioned a potential public health intervention—specifically, the importance of achieving lower blood pressure goals among older adults (lines 49-50). However, we emphasise that this is merely an illustrative example; given the wide range of conditions and associated risk factors described in our study, appropriate interventions would likely be equally diverse and require further investigation beyond the scope of this analysis.

Introduction

3. The transition from global statistics to Chile-specific challenges is abrupt, disrupting the narrative flow. Include a bridging sentence to improve coherence, such as: "While the global burden of CVD is well-documented, Chile faces unique challenges stemming from its diverse geography, aging population, and healthcare disparities."

Response: Thank you for this insightful comment. We agree that the transition into the Chile-specific context would benefit from a more explicit connection to the global narrative. In response, we have added a bridging sentence at the beginning of the paragraph to improve the coherence and flow in lines 71 to 74.

4. Repeated phrases like "leading cause of mortality". Consolidate by using synonyms, such as replacing "leading cause of mortality" with "primary contributor to death and disability," to enhance readability and avoid redundancy.

Response: Thank you for the helpful suggestion. We have revised the text to reduce redundancy by using alternative expressions where appropriate.

Methods

5. The gender-specific analysis methodology is not described adequately. It is unclear whether statistical tests were applied to validate differences.

Response: Thank you for your comment. As our study relies on descriptive analyses of age-standardised DALYs estimates provided by the GBD study, no additional statistical tests were applied to compare gender differences. Interpretations are based on visual and descriptive comparisons within the reported uncertainty intervals. This clarification has been added to the Statistical analyses section (lines 127-129).

6. Excluding third-level risk factors is not sufficiently justified, which may leave readers questioning the comprehensiveness of the study.

Response: Thank you for this comment. We have clarified in the manuscript that our decision to exclude third-level risk factors was made to improve clarity and interpretability, while focusing on the most impactful and policy-relevant contributors to the CVD burden. The revised justification can be found in lines 130-132.

7. No mention of ethical approvals or permissions for using secondary data, which is a standard requirement for publication.

Response: Thank you for the observation. We have added a statement on ethical considerations to clarify that, as the study relied exclusively on aggregated, publicly available data, no ethical approval was required. This has been included in the Ethics Considerations section (lines 130-132).

8. Include statistical tests used for gender-specific trends, e.g., "A t-test was conducted to evaluate the statistical significance of differences between gender-specific DALYs."

Response: As previously noted in our response to Comment 5, the study is based on descriptive analyses using GBD-derived estimates. No statistical tests were applied to evaluate gender-specific differences. This has been clarified in the Statistical Analyses Section (lines 127-129).

9. Justify exclusions, e.g., "Third-level risk factors were excluded to simplify analysis and focus on the most impactful contributors to CVD burden."

Response: As addressed in our response to Comment 6, we have clarified in the manuscript that third-level risk factors were excluded to enhance clarity and focus on the most impactful and policy-relevant contributors to the CVD burden. This explanation has been added to the Cardiovascular diseases and risk factors section in lines 114-116.

10. Add a note on ethical considerations, e.g., "As this study utilized publicly available secondary data, no ethical approval was required."

Response: As mentioned in our response to Comment 7, we have added a dedicated Ethics Considerations section clarifying that no ethical approval was required due to the exclusive use of aggregated, publicly available data (lines 130-132).

Results

11. Figures and tables are referenced but not thoroughly explained in the text, which may confuse readers unfamiliar with the dataset.

Response: Thank you for this helpful suggestion. We have revised the manuscript to provide clearer and more detailed explanations of the figures and tables. The modifications can be found in lines 135-138, 145-146, 157-158, 161-162, and 171-172.

12. Trends like the increase in atrial fibrillation and endocarditis are mentioned but not explored in depth.

Response: Thank you for this comment. In the Results section, we describe the trends in DALYs for endocarditis, along with other cardiovascular diseases. However, given its comparatively lower burden at the population level, we opted to focus the in-depth discussion on conditions with more substantial public health impact, such as atrial fibrillation and flutter. As noted in the discussion (lines 220-234), we provide a detailed interpretation of the atrial fibrillation trend, including its clinical consequences and potential preventive strategies.

13. Add explanations for key figures, e.g., "Figure 1 illustrates the decline in CVD DALYs over time, with the steepest reduction observed between 2000 and 2010."

Response: Thank you for the suggestion. We have added explanatory text to clarify key figures to improve interpretability for readers. These additions can be found in lines 135-138, 145-146, 157-158, 161-162, and 171-172.

14. Analyse trends more deeply, e.g., "The rise in atrial fibrillation and flutter may be attributed to increased detection rates due to advancements in diagnostic tools."

Response: Thank you for the suggestion. As noted in the Discussion section (line 223), ageing has been proposed as the pivotal risk factor for atrial fibrillation over other contributors. This is supported by evidence indicating a strong association between age and atrial fibrillation incidence, even in the absence of other comorbidities [Cheng S et al., Europace, 2024]. We chose to focus the discussion on this well-established determinant, as well as its clinical implications, including the link to cognitive decline and the need for preventive strategies.

*Reference: Cheng S, He J, Han Y, et al. Global burden of atrial fibrillation/atrial flutter and its attributable risk factors from 1990 to 2021. Europace 2024; 26. DOI:10.1093/europace/euae195.

Discussion

15. The discussion about atrial fibrillation and vascular cognitive impairment is lengthy but loosely connected to the study’s objectives.

Response: Thank you for the comment. We included this discussion to contextualise why atrial fibrillation (AF) contributes substantially to the DALYs burden, particularly in the context of an ageing population—highlighted as a pivotal risk factor for AF in our manuscript. The link with vascular cognitive impairment is clinically relevant and helps illustrate the broader consequences of AF beyond cardiovascular outcomes, thus reinforcing the importance of its prevention and management from a public health perspective.

16. Recommendations are broad and lack specificity for Chile.

Response: Thank you for the comment. As this is a descriptive study based on secondary data, our primary goal was to highlight the national burden and key drivers of CVD in Chile. While we included general recommendations based on observed trends, we now clarify in the discussion (lines 255-257) that defining specific policy actions would require additional studies incorporating local system-level and implementation considerations.

17. Focus on directly relevant implications, e.g., "The gender-specific burden of stroke and ischemic heart disease highlights the need for tailored interventions addressing unique risk profiles in men and women."

Response: Thank you for the suggestion. We agree that the gender-specific burden of stroke and ischaemic heart disease in Chile has important implications. To address this, we have added a sentence to the Interpretation and Implications section (lines 202-204) highlighting the potential value of gender-sensitive strategies in the context of primary prevention.

18. Provide concrete recommendations, e.g., "Increased funding for hypertension screening programs in rural areas of Chile could reduce DALYs attributed to high systolic blood pressure."

Response: We thank the reviewer for this suggestion. However, we consider that the concrete recommendations associated with rural areas go beyond the scope of our manuscript since we did not test rurality since analyses were performed using national-level data from the GBD Study 2021, which does not include subnational estimates for Chile. We hope a future study could address this particular gap.

Reviewer #2:

19. It is better to stratify not only by sex, but also by other very important variables; for instance; residence, occupation et. other wise the paper looks nice; on the my view it is interesting and pass to publication.

Response: Thank you for your positive feedback and thoughtful suggestions. We agree that stratification by variables such as residence or occupation would provide valuable insights. However, as this study is based on data from the Global Burden of Disease (GBD) study, such subnational or individual-level information is not available. We acknowledge this limitation in the Discussion section (lines 274-276), noting the potential relevance of regional and cultural differences within Chile.

Reviewer #3:

20. The manuscript titled "The Chilean Burden of Disability-Adjusted Life Years due to Cardiovascular Diseases: Results from the Global Burden of Disease Study 2021" is a research study that examines the impact of cardiovascular diseases (CVD) on the population of Chile over the past three decades. The study uses data from the Global Burden of Disease Study 2021 to analyze age-standardized rates of disability-adjusted life years (DALYs) due to various CVDs from 1990 to 2021, focusing on gender differences and key risk factors. The study highlights the need for continued public health efforts and multi-level interventions to address the key risk factors contributing to CVD in Chile. The authors emphasize the importance of targeted strategies to reduce the impact of high SBP, dietary risks, and other modifiable risk factors on cardiovascular health. I want to congratulate the authors on embarking on this all-important study. The methodology used in the study appears to be robust and comprehensive. However, this study has some limitations. As with any ecological research, there is a risk of ecological fallacy, where associations observed at the population level may not necessarily apply to individuals. The study notes that the measurement of low physical activity within the GBD framework is simplified, which may underestimate its impact. The study reports national-level data and does not include subnational information, which could be relevant for understanding regional differences within Chile. Overall, the methodology is well-suited to the study's objectives and thoroughly analyses the CVD burden and its risk factors in Chile.

Response: Thank you for your thoughtful summary and positive assessment of our study. We appreciate your recognition of its methodological robustness and relevance, as well as your helpful reflections on its limitations, which align with points we already acknowledged in the manuscript.

21. While the study is comprehensive and well-conducted, several aspects could be improved to enhance its robustness and applicability. Incorporating subnational data could provide more detailed insights into regional variations within Chile. This would help to identify specific areas with higher burdens of CVD and tailor public health interventions more effectively.

Response: Thank you for this suggestion. We agree that subnational analyses would offer valuable insights. However, our study is based on publicly available, national-level data from the Global Burden of Disease (GBD) Study 2021, which does not include subnational estimates for Chile. We have acknowledged this limitation in the Discussion section (lines 274-276).

22. The study mentions that the measurement of low physical activity is simplified. A more detailed analysis that considers different dimensions of physical activity (e.g., intensity, duration, type, and domain) could provide a clearer understanding of its impact on CVD.

Response: We appreciate this comment. As noted in the manuscript, physical activity is indeed a multidimensional behaviour. However, the GBD framework uses a simplified, single-indicator measure for low physical activity. Therefore, we are limited to this standardised estimate and cannot disaggregate by dimensions such as intensity or type. We have discussed this limitation in the Strengths and Limitations section (lines 267-269).

23. The study excludes third-level risk factors, such as specific dietary risks. Including these could offer a more nuanced understanding of how specific dietary components contribute to the CVD burden.

Response: Thank you for this comment. Our decision to exclude third-level risk factors was made to improve clarity and interpretability, while focusing on the most impactful and policy-relevant contributors to the CVD burden. The revised justification can be found in lines 114-116.

24. While the study examines trends over time, a more detailed longitudinal analysis could help to identify causal relationships and the impact of specific interventions over the study period.

Response: We agree that longitudinal analyses with causal inference would enrich the study. However, the GBD dataset is ecological and descriptive by nature, and does not provide individual-level data or intervention-specific tracking over time.

25. Including socioeconomic factors in the analysis could provide insights into how income, education, and access to healthcare influence the CVD burden and its risk factors.

Response: We appreciate this important observation. Unfortunately, socioeconomic variables such as income, education, and healthcare access are not included in the GBD dataset used for this study. While we acknowledge the relevance of these social determinants, their analysis would require additional data sources and a different methodological approach. We have now mentioned this limitation in the Discussion section, section limitation.

26. Incorporating qualitative data, such as patient interviews or focus groups, could provide context to the quantitative findings.

Response: Thank you for this thoughtful suggestion. We agree that qualitative data could complement quantitative findings by offering contextual insight. However, our study was limited to secondary, aggregated data from the GBD Dtudy, and did not involve primary data collection. As such, qualitative methods were beyond the scope of our research design.

27. Analyzing the impact of specific public health policies and interventions implemented in Chile over the study period could help to identify successful strategies and areas needing improvement.

Response: We agree that evaluating the impact of public health policies is essential. However, the GBD database does not include information on health policies or intervention timelines. Therefore, our study cannot link trends in CVD burden

---

## [Decision Letter · Decision Letter 1]

14 May 2025

The Chilean Burden of Disability-Adjusted Life Years due to Cardiovascular Diseases: Results from the Global Burden of Disease Study 2021

PONE-D-24-57148R1

Dear Dr. Petermann-Rocha,

We’re pleased to inform you that your manuscript has been judged scientifically suitable for publication and will be formally accepted for publication once it meets all outstanding technical requirements.

Kind regards,

Amir Hossein Behnoush

Academic Editor

PLOS ONE

Additional Editor Comments (optional):

Reviewers' comments:

Reviewer's Responses to Questions

**Comments to the Author**

1. If the authors have adequately addressed your comments raised in a previous round of review and you feel that this manuscript is now acceptable for publication, you may indicate that here to bypass the “Comments to the Author” section, enter your conflict of interest statement in the “Confidential to Editor” section, and submit your "Accept" recommendation.

Reviewer #3: All comments have been addressed

Reviewer #4: (No Response)

2. Is the manuscript technically sound, and do the data support the conclusions?

Reviewer #3: Yes

Reviewer #4: Yes

3. Has the statistical analysis been performed appropriately and rigorously? 

Reviewer #3: Yes

Reviewer #4: Yes

4. Have the authors made all data underlying the findings in their manuscript fully available?

Reviewer #3: Yes

Reviewer #4: Yes

5. Is the manuscript presented in an intelligible fashion and written in standard English?

Reviewer #3: Yes

Reviewer #4: Yes

6. Review Comments to the Author

Reviewer #3: The document is a manuscript draft for a research article analyzing the burden of disability-adjusted life years (DALYs) due to cardiovascular diseases in Chile, based on data from the Global Burden of Disease Study 2021.

The authors have adequately addressed Reviewer comments. They acknowledged the limitations of their study, which stem from the use of aggregated, national-level data from the Global Burden of Disease (GBD) Study 2021. Specifically:

1. Subnational Data: The authors agreed that incorporating subnational data would provide valuable insights but clarified that such data is not available in the GBD dataset. They acknowledged this limitation in the Discussion section

2. Physical Activity Measurement: The authors recognized the simplified measurement of physical activity in the GBD framework and discussed this limitation in the Strengths and Limitations section.

3. Exclusion of Third-Level Risk Factors: The authors justified the exclusion of third-level risk factors to improve clarity and focus on the most impactful contributors to the CVD burden.

4. Longitudinal Analysis: The authors acknowledged the value of longitudinal analyses but explained that the ecological and descriptive nature of the GBD dataset does not allow for causal inference or intervention-specific tracking.

5. Socioeconomic Factors: The authors noted the absence of socioeconomic variables in the GBD dataset and mentioned this limitation in the Discussion section.

6. Qualitative Data: The authors agreed that qualitative data could complement their findings but clarified that their study design, based on secondary data, did not involve primary data collection.

7. Public Health Policies: The authors acknowledged the importance of analyzing the impact of public health policies but explained that the GBD database does not include information on policy timelines. They encouraged future research in this area.

8. Environmental Factors: The authors recognized the importance of factors like urbanization and climate change but noted that their analysis was limited to the environmental risks reported in the GBD dataset.

Overall, the authors provided thoughtful responses to Reviewer 3's comments, acknowledged the study's limitations, and emphasized the need for future research to address these gaps.

Reviewer #4: I believe all questions and concerns have been addressed suifficiently and the manuscript is suitable for publish.

7. PLOS authors have the option to publish the peer review history of their article (what does this mean? ). If published, this will include your full peer review and any attached files.

**Do you want your identity to be public for this peer review?** For information about this choice, including consent withdrawal, please see our Privacy Policy .

Reviewer #3: **Yes: ** U S H GAMAGE

Reviewer #4: **Yes: ** Alireza Ramandi

---

## [Editor Report · Acceptance letter]

PONE-D-24-57148R1

PLOS ONE

Dear Dr. Petermann-Rocha,

I'm pleased to inform you that your manuscript has been deemed suitable for publication in PLOS ONE. Congratulations! Your manuscript is now being handed over to our production team.

Kind regards,

on behalf of

Dr. Amir Hossein Behnoush

Academic Editor

PLOS ONE